# Site Classification of *Eucalyptus urophylla* × *Eucalyptus grandis* Plantations in China

**Haifei Lu** , **Jianmin Xu \*, Guangyou Li and Wangshu Liu**

Key Laboratory of State Forestry Administration on Tropical Forestry, Research Institute of Tropical Forestry, Chinese Academy of Forestry, Guangzhou 510520, China; haifeilu1122@163.com (H.L.); luosuo3000@163.com (G.L.); 15295598916@163.com (W.L.)
\* Correspondence: jianmxu@163.com; Tel.: +86-136-0973-0753

**Abstract:** *Background and Objectives*: It is important to match species needs with site conditions for sustainable forestry. In *Eucalyptus urophylla* × *Eucalyptus grandis* plantations in southern Yunnan, China, species-site mismatches have led to inappropriate expansion and management, which has degraded forests and decreased efficiency in plantation production. Further research is needed to understand the relationship between tree growth and site productivity. We empirically explored site features and classified site types within these plantations in southern Yunnan. Our objective was to develop a theoretical basis for improving site selection for afforestation, and to establish intensive management in that region. *Materials and Methods*: 130 standard plots were set up in 1–15-year-old eucalyptus plantations in Pu'er and Lincang. We used quantification theory to examine the relationship between dominant tree growth traits and site factors. Hierarchical cluster analysis and canonical correlation analysis were applied to classify sites and evaluate the growth potential of *E. urophylla* × *E. grandis* plantations, respectively. *Results*: The multiple correlation coefficient between eight site factors (altitude, slope, slope position, aspect, soil depth, texture, bulk density, and litter thickness) and the quantitative growth of the dominant tree was 0.834 ($p < 0.05$). Slope position, altitude, and soil depth were the main factors contributing to the variation in stand growth. Plantation growth was best on lower slopes at relatively low altitude, where thick and weathered red soil layers existed. Conversely, the poorest plantations were located on upper slopes at higher altitude, with a thin semi-weathered purple soil layer. The soil factors total nitrogen (N) and potassium (K), trace boron (B), copper (Cu), and zinc (Zn) content, available phosphorous (P), and organic matter content in the soil influenced plantation growth. *Conclusions.* The addition of N, P, and K fertilizer as well as trace elements such as B, Cu, and Zn can promote the productivity of these plantations.

**Keywords:** *E. urophylla* × *E. grandis*; forest yield; plantation; silviculture; intensive management

## 1. Introduction

Eucalypts have become the most important forest tree species for industrial raw materials in southern China [1]. They are fast-growing, high-yield and widely distributed, and have a short rotation time. Increasing the area of eucalyptus plantations has become an essential way to address wood demand and meet the need for high-yield plantations [2]. Since 2004, when China outlined a plan for the national development of integrated forestry-paper production, large-scale eucalypt plantations for paper fiber have been established and in 2018 the total area of eucalypt plantations has reached 5.46 million ha nationally [3]. Yunnan is located on a plateau with a relatively low latitude, and its terrain has led to the creation of a unique forestry industry [4]. The region's existing pulpwood tree species is *Pinus kesiya* var. *langbianensis*. This species has slower growth and a longer rotation, resulting in a shortage of pulping resources. Accordingly, eucalypt plantations have developed on a large scale

in this area. However, problems such as poor stand growth and decreased production have gradually arisen because of a species-site mismatch arising from a lack of systematic site classification and site quality evaluation.

The site is the synthesis of external environmental factors in a location and the basis of forest productivity [5]. Studying the relationship between forest growth and site is of great importance for tree species selection, productivity enhancement, soil maintenance, and management [6]. Although researchers understand site quality and site selection for some tree species, there are few studies on eucalypts in China, and there is no research about site classification and site quality for eucalypts in southern Yunnan. Wu et al. found that the growth of a eucalyptus plantation on Leizhou Peninsula was affected by the combination of site productivity and soil nutrients and proposed the theory of site and nutrient effect [7]. Lai et al. used principal component analysis (PCA) to determine the dominant site factors as soil texture, slope position and slope, and used cluster analysis to zone sites for the introduction of *Eucalyptus grandis* in Sichuan into 17 types according to the soil physical and chemical factors among the three dominant factors [8]. Grant et al. analyzed the correlation between soil, climate variables and growth traits using Pearson correlation and PCA in 31 forests in north-eastern New South Wales and south-eastern Queensland and developed a site index using regression models (Available-water storage capacity and Mean annual rainfall) to improve the productivity of *E. dunnii* Maiden plantations [9].

In our study, we aimed to classify the cultivation area of eucalypt plantations and maximize the site production potential in the complex topography of southern Yunnan. We used quantification theory to examine the relationship between dominant tree growth traits and site factors. Slope position, slope angle, and soil thickness were hypothesized to be the most influential factors. We used hierarchical cluster analysis and canonical correlation analysis for site classification and evaluation. To increase the volume of tree growth, the chemical properties of soils including total N, boron (B), copper (Cu), and zinc (Zn) content, and organic matter content were investigated. Our findings contribute to the research on species-site matching and intensive management of eucalypt pulp forests in southern Yunnan, as well as provide a reference for the layout, planning, and technical innovation of eucalypt plantations in this region.

## 2. Materials and Methods

### 2.1. Study Area

The cultivation of eucalypt forests for industrial raw material in southern Yunnan covers about 60,000 ha of *E. urophylla* × *E. grandis* mainly planted by Yunnan Yun-Jing Forestry & Pulp Mill Co., Ltd. in Pu'er and Lincang. The geographical location is 22°02′−25°02′ N and 98°40′−102°19′ E (Figure 1). The planting area is in the Lancang River basin, which has a subtropical low latitude mountain monsoon climate. Because of the influence of the warm and wet Indian Ocean monsoon, there are distinct dry and wet seasons and abundant rainfall. The average annual rainfall and evaporation of this region are 1354 and 1916 mm, respectively. The rainy season extends from May to October and the average annual temperature ranges from 18 to 22 °C. The main soil types are lateritic red earth and humid-thermo ferralitic. The main vegetation species are *Ageratina adenophora* (Spreng.) R.M. King et H. Rob., *Microstegium vagans* (Nees). Camus, *Urena lobata* L. and *Phyllanthus emblica* L. The eucalypts were planted after 2000. Before that, the region had *P. kesiya* var. *langbianensis* plantations; however the slow growth and rotation time of almost 20 years, resulted in a lack of pulping resources. The company developed the wood fiber forest with *E. urophylla* × *E. grandis* DH32-16, DH32-22, and DH33-27 clones (*E. urophylla* seedlot from Alor Island, Indonesia with a selected parent pollen of *E. grandis* selected from a seedlot from Davies Creek near Mareeba in north Queensland, Australia) bred by the China-Australia Eucalypts Project in Dongmen Forest Farm, Guangxi. Measures taken during afforestation included site preparation with artificially dug holes (30 × 30 × 30 cm), and a planting spacing of 3 × 1.5 m. The base fertilizer was a special compound fertilizer for eucalypts (10:16:7, N:P:K), with an effective

nutrient value of 33%. In the year when the seedlings were planted, weeds were controlled by cutting once, and according to the degree of weed growth, weeding was carried out before the wet season. The following year, the weeds were controlled by cutting twice, and topdressing was done once at 0.5 kg per plant. In the third year, the weeds were controlled by cutting once, and topdressing was done once at 0.7 kg per plant. The compound fertilizer for top dressing N:P:K ratio was 18:12:10.

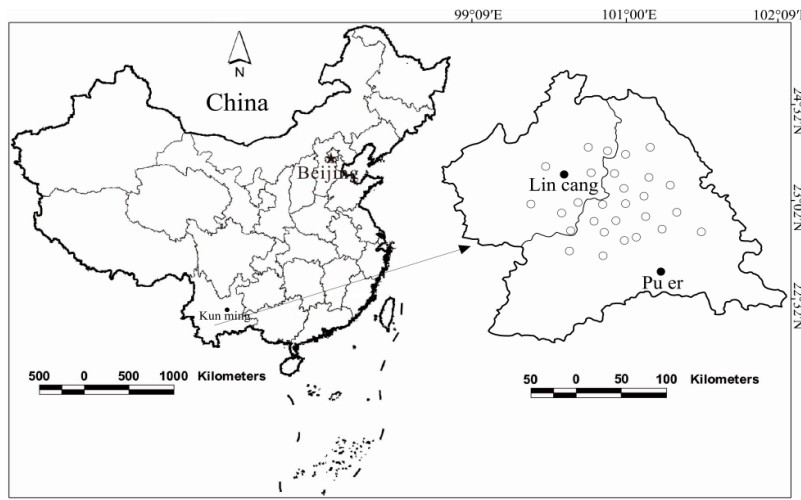

**Figure 1.** Location of study sites (white circles) in southern Yunnan, China.

### 2.2. Experimental Design and Data Compilation

*E. urophylla* × *E. grandis* plantations, aged between 1 and 15 years, served as the investigation objects. During November and December 2017, we employed a typical sampling method to select a total of 130 plots (20 m × 20 m) along the forest roads of the management area. The selection procedure was based on variations of site factors, including altitude, slope angle, slope position, slope aspect, and soil type (Table 1). For each variation, we selected a minimum of 2 plots.

**Table 1.** Category classification of site factors.

| Item | Category | | | | |
|---|---|---|---|---|---|
| Altitude (m) $X_1$ | <1000 | 1000−1200 | 1200−1400 | 1400−1600 | 1600−1800 |
| Slope (°) $X_2$ | Conservative slope 0−5 | Gentle slope 6−15 | Incline slope 16−25 | Steep slope >25 | |
| Slope position $X_3$ | Upper slope | Middle slope | Lower slope | Flat ground | |
| Soil depth (cm) $X_4$ | Thin: <40 | Medium: 40−80 | | Thick: >80 | |
| Slope aspect $X_5$ | Sunny slope | Half-sunny slope | Half-shady slope | Shady slope | Flat slope |
| Texture $X_6$ | Sandy loam | Light loam | Middle loam | Heavy loam | |
| Bulk density (g·cm$^{-3}$) $X_7$ | >1.4 | 1.2−1.4 | 1.0−1.2 | <1.0 | |
| Litter thickness (cm) $X_8$ | Thin: <5.0 | Medium: 5.0−10.0 | | Thick: >10.0 | |

The geographic coordinates and altitude were recorded using GPS (Garmin GPSMAP60CSX, SHH, China), and all growth traits of *E. urophylla* × *E. grandis* were measured in each sample plot. The soil samples were collected by soil auger in each plot, and three soil profiles were excavated at equal distance in a diagonal direction with the depth of 0–60 cm and 80 cm wide. The soil samples in each plot were mixed into a composite sample, weighing approximately 1 kg.

The statistics of understory shrubs were recorded by setting five subplots with an area of $2 \times 2$ m in the four corners and center of the sample plot. At each standard plot, 5 dominant trees were selected and their average height was taken as the dominant tree height [10]. Furthermore, one dominant tree was cut for stem analysis every 2 m. In this way, the dominant tree height at the base age was obtained as the site index by reading the disc.

In addition, five core wood samples with a size of 5 mm were taken at breast height from trees of an average diameter from the north to south side for testing wood properties. The sample was taken by removing the two ends of the trunk, where they were cut off 3 m from the top and 1 m from the base. The samples were then chopped into 3–4 cm chips, and 5.0 kg were taken for pulping testing.

The quantitative regression model formula of *E. urophylla* × *E. grandis* is as follows [11,12]:

According to *m* items, $(X_1, \ldots, X_m)$ predict a certain variable *y*. It is assumed that the projected $X_j$ has $r_j$ categories, and observation data for *n* samples are obtained. Before we calculate this, some notations will be introduced. Let

$$\delta_i(j,k) \begin{cases} =1 & \text{if } i\text{-person checks in the } k\text{-th sub-category in the } j\text{-th item,} \\ =0 & \text{otherwise.} \end{cases} \quad (1)$$
$$(i = 1, \ldots, n; \quad 1, \ldots, m; \quad k = 1, \ldots, r_j)$$

This is the reaction of category *k* of item *j* in the *i* sample. In this way, the original data is the response matrix:

$$X = \begin{pmatrix} \delta_1(1,1) & \cdots & \delta_1(1,r_1) & \cdots & \delta_1(m,1) & \cdots & \delta_1(m,r_m) \\ \delta_2(1,1) & \cdots & \delta_2(1,r_1) & \cdots & \delta_2(m,1) & \cdots & \delta_2(m,r_m) \\ \vdots & & \vdots & & \vdots & & \vdots \\ \delta_n(1,1) & \cdots & \delta_n(1,r_1) & \cdots & \delta_n(m,1) & \cdots & \delta_n(m,r_m) \end{pmatrix} \text{ and } y_i = (y_1, \ldots, y_n), \quad (2)$$

where $y_i$ is the observed value of the variable *y* in the *i* sample.

$$y_i = \sum_{j=1}^{m} \sum_{k=1}^{r_j} \delta_i(j,k)b_{jk} + \varepsilon_i, (i = 1, ..., n) \quad (3)$$

where *y* is the dependent variable. $\delta_i (j, k)$ is the score of category *k* of item *j*. $b_{jk}$ is an unknown coefficient and $\varepsilon_i$ is a random error. The least squares estimate $b_{jk}$ is the reaction value (0 or 1) of category *k* of item *j*. *m* is the number of factors, that is, the site factors fitted by each parameter, such as altitude, slope, etc. *n* is the number of grades, *k* is the category divided under factor *j* (item). The prediction equation of *E. urophylla* × *E. grandis* is as follows:

$$\widehat{y} = \sum_{j=1}^{m} \sum_{k=1}^{r_j} \delta_i(j,k)\widehat{b}_{jk} \quad (4)$$

Redundancy analysis (RDA) is a constrained principal component analysis ranking method that can evaluate the relationship between one or a group of variables and another group of multivariate data from a statistical point of view [13]. The rough arrows represent biological factors, and the dotted lines represent environmental factors. The angle between them represents their correlation [14].

The single timber volume of dominant trees was calculated as:

$$V = H_1 \times \pi \times (r_1{}^2 + r_1 \times r_2 + r_2{}^2)/3 + H_2 \times \pi \times (r_2{}^2 + r_2 \times r_3 + r_3{}^2)/3 + \ldots\ldots + H_{n-1} \times \pi \quad (5)$$
$$\times (r_{n-1}{}^2 + r_{n-1} \times r_n + r_n{}^2)/3 \quad (n > 2)$$

where V represents the volume ($m^3$), H is the length of each piece of wood, $r_{n-1}$ and $r_n$ are the lower radius and upper radius of the wood segment, respectively.

### 2.3. Sample Soil Analysis

For soil analysis, 11 variables were assessed including pH, organic matter, total N, P, K, available N, P, K, B, Zn, and Cu [15]. Soil pH was measured by the potentiometric method at a soil/water ratio of 1:2.5 (10 g of soil and 25 mL of $H_2O$). Soil organic matter was measured by the potassium dichromate oxidation-capacity method, where 0.3 g of soil was boiled for 4.3 min with 5 mL of $K_2Cr_2O_7$ and 5 mL of $H_2SO_4$, titrated with $FeSO_4$. Total N was measured by the alkaline hydrolysis diffusion method. A sample of 0.5 g of soil was digested with 1 g catalyst and 5 mL of $H_2SO_4$ then diffused with 10 mL of 1.8 mol $L^{-1}$ NaOH and titrated with 0.01 mol $L^{-1}$ HCl. Total P and total K were measured by the sodium hydroxide alkali melting-molybdenum antimony colorimetric method. A sample of 0.25 g of soil was boiled with 2 g of NaOH for 15 min and, after filtering, A700 spectrophotometry was used for colorimetry. Available N was measured by the alkaline hydrolysis diffusion method as well but using $FeSO_4$ and boric acid to digest. Available P was extracted by a 50 mL mixture of 0.05 mol $L^{-1}$ HCl and 0.025 mol $L^{-1}$ $H_2SO_4$. B was measured by curcumin colorimetry method. Cu and Zn were measured by atomic absorption spectrophotometry method. The samples were extracted with 50 mL of 0.1 mol $L^{-1}$ HCl by shaking for 1.5 h.

### 2.4. Data Analysis

Quantification theory was used to list each site factor by (0, 1) and SPSS 13.0 for Windows (SPSS Inc., New York, NY, USA) was used for quantitative regression analysis. The hierarchical cluster analysis (HCA) of the leading site factors was calculated by SPSS. In addition, DCA and RDA of the dominant tree height and DBH relative to soil chemical properties was performed in R 3.6.1 (Lucent Technologies, Trenton, NJ, USA). Soil physical and chemical properties and the productivity grade trends of *E. urophylla × E. grandis* were plotted in Excel 2016 (Microsoft, Redmond, WA, USA).

## 3. Results

### 3.1. Determination of the Base Age

The base age refers to the age at which the growth of the dominant tree reaches peak or tends to stabilize. Generally, the base age of fast-growing tree species is relatively low, and that of slow-growing tree species is relatively high [10]. It can be seen from Table 2 that the coefficient of variation of tree height was large before 5 years, and then tree height growth tended to stabilize and did not show significant differences by the 6th year. Therefore, the base age of *E. urophylla × E. grandis* was 6 years. The dominant trees at the sixth years (base age) of 85 sample plots were selected for site classification and the remaining 45 sample plots were used to study the growth and development process of *E. urophylla × E. grandis* plantations.

**Table 2.** Coefficient of variation of dominant height in different age classes.

| Age Class | 1 | 2 | 3 | 4 | 5 | 6 | 7 | 8 | 9 | 10 | 11 | 12 | 13 | 14 | 15 |
|---|---|---|---|---|---|---|---|---|---|---|---|---|---|---|---|
| SD of height | 1.71 | 2.24 | 2.43 | 2.67 | 2.09 | 2.34 | 2.39 | 2.43 | 2.48 | 2.52 | 2.59 | 2.61 | 2.66 | 2.68 | 2.70 |
| Average height/m | 6.4f | 10.6e | 14.6d | 17.7c | 21.1b | 23.8a | 24.2a | 24.6a | 25.1a | 25.5a | 26.1a | 26.5a | 26.8a | 27.1a | 27.5a |
| CV of height/% | 26.72 | 21.13 | 16.64 | 15.08 | 9.91 | 9.83 | 9.87 | 9.92 | 9.88 | 9.96 | 9.92 | 9.85 | 9.93 | 9.89 | 9.82 |

Note: SD, standard deviation; CV, coefficient of variance. The value with the same letters have not significant difference ($p = 0.05$).

### 3.2. Formulating a Quantitative Regression Model

The site factors and dominant height are listed in Table S1. By combining site factor classification with standard survey information (Table 1), each site factor in the plots was listed in (0, 1). Quantitative

theory was used for site index modeling, and the site factor categories were obtained by fitting regression coefficients and related parameters of height growth of the dominant tree species, i.e., *E. urophylla × E. grandis* (Table 3).

**Table 3.** Quantity regression result of site factors.

| Item | Category | Code | Score | Score Range and Proportion | Coefficient of Partial Correlation |
|---|---|---|---|---|---|
| Constant | | | 27.669 | | |
| Altitude (m) $X_1$ | <1000 | $X_{11}$ | 0 | 3.455 (21.07%) | 0.362 ** |
| | 1000−1200 | $X_{12}$ | −1.513 | | |
| | 1200−1400 | $X_{13}$ | −2.170 | | |
| | 1400−1600 | $X_{14}$ | −2.827 | | |
| | 1600−1800 | $X_{15}$ | −3.455 | | |
| Slope $X_2$ | Conservative Slope | $X_{21}$ | 1.421 | 2.392 (14.59%) | 0.165 * |
| | Gentle slope | $X_{22}$ | 0.677 | | |
| | Incline slope | $X_{23}$ | 0 | | |
| | Steep slope | $X_{24}$ | −0.971 | | |
| Slope position $X_3$ | Flat ground | $X_{31}$ | −1.672 | 3.713 (22.65%) | 0.437 ** |
| | Upper slope | $X_{32}$ | 0 | | |
| | Middle slope | $X_{33}$ | 0.856 | | |
| | Lower slope | $X_{34}$ | 2.041 | | |
| Soil depth (cm) $X_4$ | Thin: <40 | $X_{41}$ | −1.189 | 2.704 (16.49%) | 0.232 * |
| | Medium: 40−80 | $X_{42}$ | 0 | | |
| | Thick: >80 | $X_{43}$ | 1.515 | | |
| Slope aspect $X_5$ | Sunny slope | $X_{51}$ | −1.179 | 1.729 (10.55%) | 0.184 * |
| | Half-sunny slope | $X_{52}$ | 0 | | |
| | Half-shady slope | $X_{53}$ | 0.141 | | |
| | Shady slope | $X_{54}$ | 0.550 | | |
| | Flat slope | $X_{55}$ | 0 | | |
| Texture $X_6$ | Sandy loam | $X_{61}$ | −0.009 | 0.671 (4.09%) | 0.279 * |
| | Light loam | $X_{62}$ | 0.608 | | |
| | Middle loam | $X_{63}$ | 0 | | |
| | Heavy loam | $X_{64}$ | −0.063 | | |
| Soil bulk density (g·cm$^{-3}$) $X_7$ | >1.4 | $X_{71}$ | −2.231 | 2.231 (13.79%) | 0.307 * |
| | 1.2−1.4 | $X_{72}$ | −2.153 | | |
| | 1.0−1.2 | $X_{73}$ | −2.084 | | |
| | <1.0 | $X_{74}$ | −2.053 | | |
| Litter thickness (cm) $X_8$ | Thin: <5.0 | $X_{81}$ | −1.135 | 1.863 (11.36%) | 0.232 * |
| | Medium: 5.0−10.0 | $X_{82}$ | 0 | | |
| | Thick: >10.0 | $X_{83}$ | 0.728 | | |

Note: ** Statistical significance at $p < 0.01$, * Statistical significance at $p < 0.05$.

The quantitative regression equation was as follows: $Y = 27.669 − 1.513 X_{12} − 2.170 X_{13} − 2.827 X_{14} − 3.455 X_{15} + 1.421 X_{21} + 0.677 X_{22} − 0.971 X_{24} − 1.672 X_{31} + 0.856 X_{33} + 2.041 X_{34} − 1.672 X_{34} − 1.189 X_{41} + 1.515 X_{43} − 1.179 X_{51} + 0.141 X_{53} + 0.550 X_{54} − 0.009 X_{61} + 0.608 X_{62} − 0.063 X_{64} − 2.231 X_{71} − 2.153 X_{72} − 2.084 X_{73} − 2.053 X_{74} − 1.135 X_{81} + 0.728 X_{83}$.

The following site factors were the most important in the model (in order most to least important): slope position (0.2265), altitude (0.2107), soil depth (0.1649), slope angle (0.1459), soil bulk density (0.1379), litter thickness (0.1136), slope aspect (0.1055), texture (0.0409). The first three factors jointly

explained 60.21% of the total variation in tree growth. Slope position and altitude affected the growth of *E. urophylla* × *E. grandis* plantations more than slope angle and slope aspect. A negative value for flat slope indicated that growth was poor, while the coefficient value increased from the upper slope to lower slope, indicating that the growth trend was better. The higher the altitude, the lower the coefficient value, indicating that growth was worse. The coefficient of soil thickness increased from thin to thick, indicating that the thicker the soil depth, the better the growth. The results of the T-test showed that the partial correlation between the eight site factors was significant ($p < 0.05$). The complex correlation coefficient was R = 0.834, and that was statistically significant ($F_{(8,76)} = 17.412 > F_{0.05\ (8,76)} = 2.063$). Thus, the eight selected site factors explained 83.4% of tree growth in these plantations.

### 3.3. Site Classification and Evaluation

According to the HCA, the most influential factors on the dominant height were slope position, altitude, and soil depth. When the relative distance was 5 (Figure S1), the 85 sample plots were clustered into thirteen different site types based on the difference in slope, altitude, and soil depth (Table 4).

**Table 4.** Site classification of *E. urophylla* × *E. grandis* plantation in southern Yunnan.

| Basis of Classification | | | Site Type | Dominant Height/m | Range of Change/m |
|---|---|---|---|---|---|
| Slope Position | Altitude/m | Soil Depth/cm | | | |
| Upper slope | 1000–1200 | >40 | Upper slope, medium-low altitude, medium-thick soil | 24.2 | 21.9–25.3 |
| | 1000–1200 | <40 | Upper slope, medium-low altitude, Thin soil | 19.2 | 18.5–21.7 |
| | 1200–1400 | * | Upper slope, medium altitude, total soil thickness | 23.6 | 21.9–25.3 |
| | 1400–1800 | * | Upper slope, medium-high altitude, total soil thickness | 21.5 | 15.8–23.4 |
| Middle slope | 1000–1200 | >40 | Middle slope, medium-low altitude, medium-thick soil | 22.9 | 19.5–24.5 |
| | 1000–1200 | <40 | Middle slope, medium-low altitude, Thin soil | 18.5 | 18.1–18.9 |
| | 1200–1400 | >40 | Middle slope, medium altitude, medium-thick soil | 22.4 | 20.5–25.4 |
| | 1400–1800 | >40 | Middle slope, medium-high altitude, medium-thick soil | 21.8 | 20.8–23.0 |
| Lower slope | 900–1000 | >40 | Lower slope, low altitude, medium-thick soil | 25.4 | 24.5–26.2 |
| | 1000–1200 | >80 | Lower slope, medium-low altitude, thick soil | 24.8 | 23.2–25.5 |
| | 1200–1400 | >40 | Lower slope, medium altitude, medium-thick soil | 21.1 | 20.1–22.3 |
| | 1400–1600 | 40–80 | Lower slope, medium-high altitude, medium soil | 19.5 | 19.0–20.0 |
| Flat slope | 1400–1600 | 40–80 | Flat slope, medium-high altitude, medium soil | 20.1 | 19.8–24.5 |

Note: * indicates that the soil layer covers three levels of thickness.

Table 5 shows the three productivity grades of the site: high- (24–27 m), medium- (21–24 m) and low-yield (18–21 m), which were obtained from the differences in dominant height. The evaluation was carried out with reference to the orthostatic factor regression model and followed the principle that the predicted values of the model were close to the measured values. According to Table 5, lower slopes and altitudes, and thicker soil layers meant better growth of the trees.

**Table 5.** Site evaluation of *E. urophylla* × *E. grandis* plantation in southern Yunnan.

| Site Productivity Groups | Site Type | | Dominant Height Predicted/m | Dominant Height Measured/m |
|---|---|---|---|---|
| High-yield group I | I$_1$ | Lower slope, 900−1000 m, thick soil | 25.6 | 25.4 |
| | I$_2$ | Lower slope, 1000−1200 m, thick soil | 25.2 | 24.8 |
| | I$_3$ | Upper slope, 1000−1200 m, medium-thick soil | 24.7 | 24.2 |
| Middle-yield group II | II$_1$ | Upper slope, 1200−1400 m, total soil depth | 24.1 | 23.6 |
| | II$_2$ | Middle slope, 1000−1200 m, medium-thick soil | 23.5 | 22.9 |
| | II$_3$ | Middle slope, 1200−1400 m, medium-thick soil | 22.7 | 22.4 |
| | II$_4$ | Middle slope, 1400−1800 m, medium-thick soil | 22.2 | 21.8 |
| | II$_5$ | Upper slope, 1400−1800 m, total soil depth | 21.4 | 21.5 |
| | II$_6$ | Lower slope, 1200−1400 m, medium-thick soil | 20.9 | 21.1 |
| Low-yield group III | III$_1$ | Flat ground, 1400−1600 m, medium soil | 20.4 | 20.1 |
| | III$_2$ | Lower slope, 1400−1600m, medium soil | 19.3 | 19.5 |
| | III$_3$ | Upper slope, 1000−1200 m, thin soil | 18.9 | 19.2 |
| | III$_4$ | Middle slope, 1000−1200 m, thin soil | 18.3 | 18.5 |

To be specific, the high-yield group was mainly located on lower slopes, where the soil depth was thick or medium-thick and the altitude was from 1000 to 1200 m. The height growth, DBH, and mean annual increment of *E. urophylla* × *E. grandis* were 24.8 m, 14.9 cm, and 38.47 m$^3$·ha$^{-1}$·a$^{-1}$, respectively.

The middle-yield group was mainly located on the middle and upper slopes, where the soil depth was medium-thick, and the altitude ranged from 1000 to 1800 m. The height growth, DBH, and mean annual increment of *E. urophylla* × *E. grandis* were 22.2 m, 13.8 cm, and 27.34 m$^3$·ha$^{-1}$·a$^{-1}$, respectively.

The low-yield group was mainly located on the upper slope, where the soil depth was thin, and the altitude was from 1000 to 1600 m. The height growth, DBH, and mean annual increment of *E. urophylla* × *E. grandis* were 18.3 m, 12.3 cm, and 15.67 m$^3$·ha$^{-1}$·a$^{-1}$, respectively.

### 3.4. Relationship between the Growth of E. urophylla × E. grandis and Soil Chemical Properties

The Detrended Correspondence Analysis (DCA) of growth traits of *E. urophylla* × *E. grandis* showed that the length of gradient of first axis was 2.73, which was less than 3, indicating that the data was suitable for RDA. RDA significantly revealed the response of *E. urophylla* × *E. grandis* growth traits to soil factors. As shown in Table 6, the eigenvalues of axis 1 and axis 2 were 0.511 and 0.035, respectively, and the cumulative interpretation percentage was 73.7%, which could reflect the relationship between growth traits and soil factors.

**Table 6.** RDA ranking of trait and soil chemical property of *E. urophylla* × *E. grandis*

| Item | Axis 1 | Axis 2 | Axis 3 | Axis 4 | |
|---|---|---|---|---|---|
| Eigenvalues | 0.511 | 0.035 | 0.004 | 0.000 | |
| Correlation of trait-soil factors | 0.872 | 0.548 | 0.533 | 0.475 | |
| Cumulative variation of trait /% | 71.3 | 73.7 | 74.1 | 74.2 | |
| Sum of all eigenvalues | | | | | 1 |
| Sum of canonical eigenvalues | | | | | 0.551 |
| Permutation test on first axis (*F* test) | *F* = 28.65 | *p* = 0.002 | | | |
| Permutation test on all axis (*F* test) | *F* = 7.832 | *p* = 0.002 | | | |

Axis 1 mainly reflected the effects of soil total N, available B, P, Zn, Cu content, soil organic matter content and total K on the growth characteristics of *E. urophylla × E. grandis*. Axis 2 mainly reflected the effects of soil pH, available K, total P and available N on growth traits. From the ranking chart of RDA, it can be seen that the response of growth traits of *E. urophylla × E. grandis* to soil factors was different and so was the degree. The height and DBH of *E. urophylla × E. grandis* had strong positive correlation with total N, available B, P, Zn, Cu content, soil organic matter content, and total K content, while had negative correlation with soil pH (Figure 2). In this way, soil total N, available B, P, Zn, Cu content, soil organic matter content and total K content are the key soil factors affecting the growth characteristics of *E. urophylla × E. grandis*.

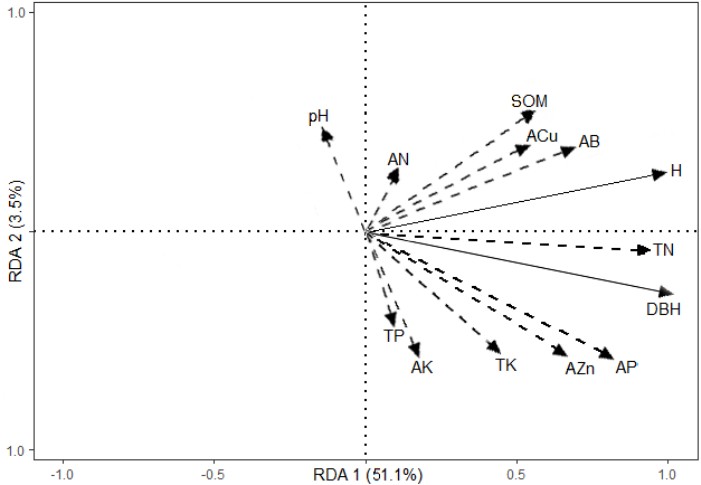

**Figure 2.** RDA of soil chemical property and trait of *E. urophylla × E. grandis.* Height of dominant trees (H); Diameter at breast height (DBH); Soil pH value (pH); Soil organic matter content (SOM); Soil total nitrogen (TN); Soil total phosphorus (TP); Soil total potassium (TK); Soil available nitrogen (AN); Soil available phosphorus (AP); Soil available potassium (AK); Available Cu (ACu); Available Zn (AZn); Available boron (AB).

As can be seen in Figure 3, through the analysis of all soil chemical factors and *E. urophylla × E. grandis* productivity, the total N content of soil in the high-yield group was 47.7% and 81.1% higher than that in the middle- and low-yield group, respectively. The soil organic matter content in the high- and middle-yield groups was slightly different, but it was 49.8% and 65.1% higher than that in the low-yield group, respectively. The available P content in the high-yield group was 66.7% and 41.7% higher than that in the middle- and low-yield groups, respectively. The total P content of the soil in the high- and middle-yield groups was slightly different, but it was 45.8% and 47.9% higher than that in the low-yield group, respectively. The total K content in the middle-yield group was 5.6% and 72.7% higher than that in the high- and low-yield groups, respectively. The available K content in the high-yield group was 49.5% and 23.2% higher than that in the middle- and low-yield groups, respectively. The effective B content of the high-yield group was 33.3% and 86.7% higher than that of the middle- and low-yield groups, respectively. The effective Zn content of the high-yield group was 21.6% and 63.6% higher than that in the middle- and low-yield groups, respectively. The effective Cu content of the high-yield group was 22.9% and 40.9% higher than that of the middle- and low-yield groups, respectively. The high-yield group was mainly located on the lower slope, where the average annual growth was better because the climate was warmer and wetter at lower attitude, and the thicker soil had a higher nutrient content. In contrast, the low-yield group was mainly located on the higher altitude upper slope, where the average annual growth was smaller because there was less water and heat, and the thin soil had a low nutrient content.

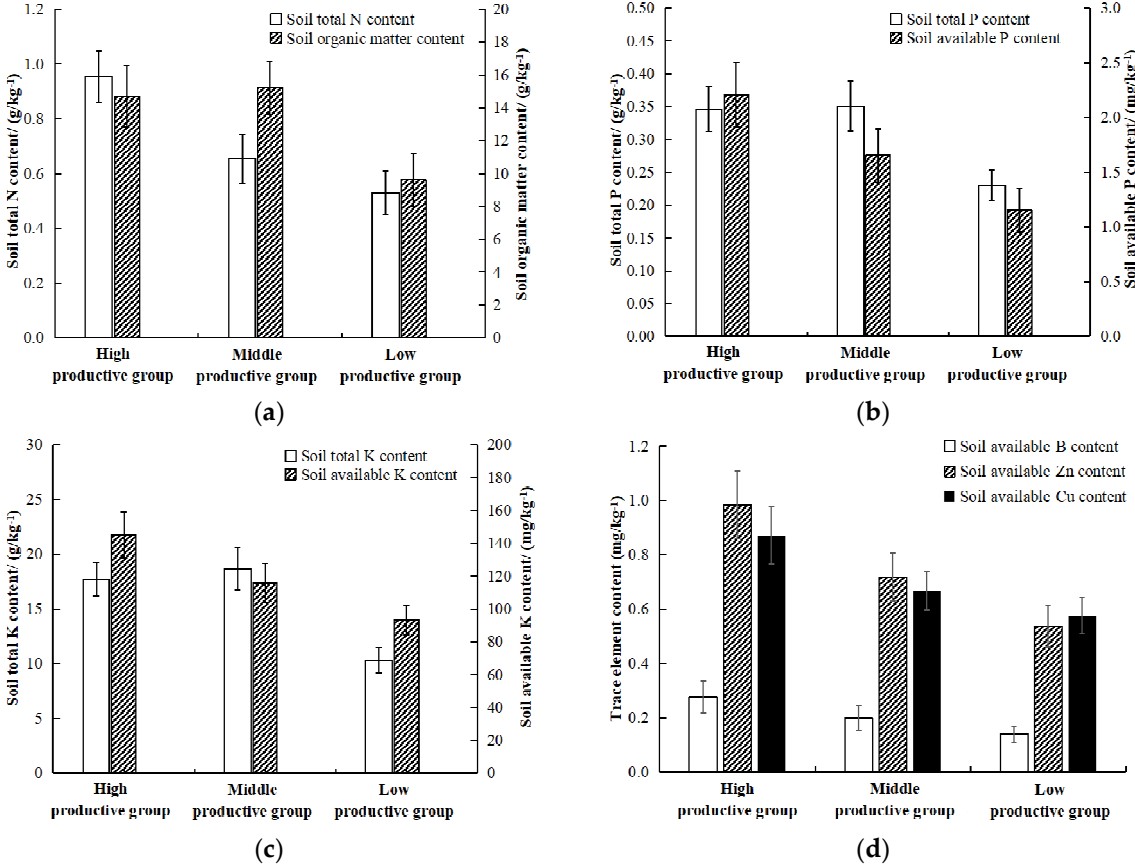

**Figure 3.** Main chemical and physical properties for each productivity group. (**a**) Performance of soil total N content and soil organic matter content under different productivity group, (**b**) Performance of soil total P content and soil available P content under different productivity group, (**c**) Performance of soil total K content and soil available K content under different productivity group, (**d**) Performance of soil available B content, soil available Zn content and soil available Cu content under different productivity group.

## 4. Discussion

*4.1. Differences between Site Types of E. urophylla × E. grandis in Southern Yunnan and Other Regions*

The Leizhou Forestry Bureau has more than 70 years of eucalypt cultivation history, with several varieties planted such as *E. exserta*, *E. leizhou* No.1, *E. urophylla*, *Eucalyptus ABL* No.12, *E. urophylla* U6 clone, and new varieties developed in recent years such as LH4, LH6, LH22 and LH28 clones selecting from hybrids of TH9211 (*E. urophylla × E. camaldulensis*) and TH9117 (*E. urophylla × E. tereticornis*) bred by the Research Institute of Tropical Forestry, CAF, etc. Zhang et al. conducted a site index model study on *Eucalyptus ABL* No. 12 (W5 clone) [16]. Zhen et al. adopted successive generations of continuous crop farming and found that soil organic matter decreased significantly [17,18]. The average single plant biomass decreased with an increase of continuous cropping, and fertility declined [19]. At the same time, frequent summer typhoons in the coastal areas of southern China were a major threat to the growth and cultivation of eucalypts [20]. The survey period was relatively short, approximately 4 to 5 years, so it was difficult to classify site types in the Leizhou Peninsula. Since the 1990s, eucalypt cultivation in southern Yunnan has developed rapidly [21]. Since 2001, Yunnan Yunjing Forestry and Pulp Mill Co., Ltd. and Leizhou Forestry Bureau have carried out joint introductions and large-scale planting of DH clones of *E. urophylla × E. grandis*. The site status in this paper is quite different as the reforestation in southern Yunnan has not yet begun, the stand of the raw forest belongs to the first generation forest which does not lead to the decline of soil fertility because of continuous reafforestation

and regenerations by crops. In addition, there were no meteorological factors such as typhoons in this area during the study period, and the study sample plots were mainly from 6 to 15 years old, which better reflected the site productivity of forest land. Together, the best clones represented by *E. urophylla* × *E. grandis* DH clones were screened out, and their average annual accumulated growth was more than 30 m$^3$ ha$^{-1}$, which concurred with the site production potential of this study [22]. Hence, our research was able to develop a site classification of *E. urophylla* × *E. grandis* plantation in southern Yunnan.

### 4.2. Main Impact Factors on Site Types of E. urophylla × E. grandis in Southern Yunnan

Site classification and evaluation are the basis of scientific afforestation methods and are particularly important for evaluating the potential of site production [23]. There has been some related research into the factors for site classification and evaluation. For example, slope position, altitude, and slope aspect were the dominant factors for site classification of *Betula alnoides* plantations in the southwestern Pingxiang, Guangxi [24]. Soil nutrients, density, and texture were the dominant factors for the site types of *Tectona grandis* plantations in Xishuangbanna, Yunnan [25]. In this paper, slope position, altitude, and soil depth were the dominant factors for the site types of *E. urophylla* × *E. grandis* plantations in southern Yunnan. Moreover, we found that the growth of eucalypts was also related to the soil physical and chemical properties on different slopes. To be specific, the soil on the lower slope was relatively looser than that of the middle and upper slope. The aeration, water permeability, water conservation and fertilizer retention ability on the lower slope were better than those of upper slope. All these factors facilitated root growth and normal development [26]. With the decrease of the slope, the soil depth and soil moisture content were larger, which was related to the physical and chemical properties of the soil, and had a great impact on tree growth [27].

Additionally, some studies have shown that the biomass of eucalypt plantations is significantly negatively correlated with the increase of altitude, which changes the hydrothermal conditions, affecting the growth and development of stands, and reducing the biomass [28–31]. In this study, the altitude of *E. urophylla* × *E. grandis* in the high-yield group was generally below 1400 m, while the altitude of the middle- and low-yield groups was more than 1400 m. Therefore, we also found that at higher altitudes, sites were less suitable for the growth of *E. urophylla* × *E. grandis*.

### 4.3. Physical and Chemical Properties of Soil and Productivity

Nitrogen fertilizer can increase the plant chlorophyll content and photosynthetic rate and can promote the growth of plant stems and leaves [32]. B, Zn, and Cu are essential trace elements for plant growth and development. A lack of these three trace elements is extremely detrimental to the growth of young eucalypt forests because it inhibits growth and causes physiological diseases [33–35]. When elements such as B, Zn, Cu, and Mn were applied, the production of *Eucalyptus ABL* No. 12 (W5 clone) and *E. urophylla* exceeded that of unfertilized sites within three years [36]. B participates in the metabolism and transportation of carbohydrates in plants and promotes the growth and development of the root system. B deficiency will lead to thin, curly, dark, and dull leaves, and the tip growth will be inhibited, or withered shoots and branches will be produced. A lack of Zn results in shortened node spacing, smaller leaves, clusters of tender shoots, reddish leaves and leaf edges, or withered leaves, while Cu deficiency causes deformed leaves and shoots [37].

Soil organic matter content represents soil fertility and is an important source of various nutrients such as N, P, and trace elements required by plants. Increasing organic matter content enables a high and stable yield of eucalypts, and thus forest litter should be kept as much as possible [38]. Total N in soil has a great impact on the productivity level of *E. urophylla* × *E. grandis*, and ethanol deficiency can inhibit the growth of above-ground parts and roots of eucalypts, affect chlorophyll synthesis, and cause leaf yellowing as well as spot blight [39].

Our results were performed according to the soil nutrient grading standards of the second National Soil Census (Table 7) [40]. The ranges of average content of soil organic matter and total N in our

findings were 9.61–15.87 and 0.53–0.96 g kg$^{-1}$ respectively, both of which can be regarded as Level IV or V. Available P ranged from 1.21 to 2.04 mg kg$^{-1}$, belonging to Level VI. Available K ranged from 95.3 to 142.1 mg kg$^{-1}$, belonging to Level III or IV. Available B ranged from 0.15 to 0.28 mg kg$^{-1}$, available Zn ranged from 0.55 to 0.91 mg kg$^{-1}$, and available Cu ranged from 0.61 to 0.86 mg kg$^{-1}$. Therefore, for medium- and low-yield groups, N and P fertilizer should be appropriately added, along with trace elements such as B, Zn, and Cu, to promote the growth and development of forests. The logging operations should retain the dead branches, fallen leaves, and harvested residues in the forestland to increase soil organic matter, thus providing a resource for later nutrient recycling.

**Table 7.** Standard for nutrient classification of the second National Soil Census.

| Soil Chemical Property | | Level | | | | | |
|---|---|---|---|---|---|---|---|
| | | I | II | III | IV | V | VI |
| Organic matter | g/kg | >40 | 30–40 | 20–30 | 10–20 | 6–10 | <6 |
| Total N | g/kg | >2.0 | 1.5–2 | 1–1.5 | 0.75–1 | 0.5–0.75 | <0.5 |
| Available P | mg/kg | >40 | 20–40 | 10–20 | 5–10 | 3–5 | <3 |
| Available K | mg/kg | >200 | 150–200 | 100–150 | 50–100 | 30–50 | <30 |
| B | mg/kg | >2.0 | 1.0–2.0 | 0.5–1.0 | 0.2–0.5 | <0.2 | |
| Zn | mg/kg | >3.0 | 1.0–3.0 | 0.5–1.0 | 0.3–0.5 | <0.3 | |
| Cu | mg/kg | >1.8 | 1.0–1.8 | 0.2–1.0 | 0.1–0.2 | <0.1 | |

Note: I represents extremely high level; II represents high level; III represents upper middle level; IV represents lower middle level; V represents low level; VI represents extremely low level. (Soil types in China, 1996).

Additionally, the soil of high-yield group was composed of red soil developed by the weathering of argillaceous rocks, which was intact and deep, ensuring long-lasting and vigorous tree growth. Therefore, sites with this kind of soil were the best for cultivation of *E. urophylla* × *E. grandis* with high pulpwood productivity. The soil of middle-yield group was mainly composed of lateritic red soil and natural red soil with deep development, i.e., good site conditions and production potential. Therefore, these were also ideal for cultivating *E. urophylla* × *E. grandis* with high pulpwood under proper management. The soil of low-yield group was mainly composed of purple sandy shale semi-weathered purple soil, where biological accumulation was weak, soil development was shallow, water retention ability was low, and the site condition was poor. Thus, these sites were not suitable for the cultivation of fast-growing and high-yielding eucalypts. However, they could be used to cultivate native tree species such as *P. kesiya* var. *langbianensis*, so as to avoid economic losses.

### 4.4. Implications and Recommendations for Forest Management

It is well known that eucalypts grow rapidly, but inevitably reduce soil fertility in the process, which means that the maintenance of soil fertility is particularly important. Therefore, when cutting trees that then regenerate from sprouts, the harvested residues, roots, and eucalypts leaves that cannot be used as raw material should be retained on the forest land. When site preparation and subsequent tending of young plantations are carried out, measures such as the rotation of legumes and non-legumes may maintain and restore soil fertility [41]. It is beneficial to contribute to maintaining the stocks of nutrients in the soils cultivated with eucalypts [42]. According to research on the growth period of eucalypt plantations in Brazil, it was found that even though eucalypts were planted in soils with low natural fertility and low available P, the trees showed a high growth rate when the nutrient management was adequate, such as adding appropriate P fertilizers [43–45].

Moreover, the growth of *E. urophylla* × *E. grandis* in southern Yunnan was greatly affected by the content of N, B, Zn and organic matter in the soil. In order to meet the normal growth requirements, it is essential to create a benign ecosystem with a suitable water and nutrient balance in the stands. For example, N, P, K compound fertilizer, trace elements, chicken manure and inter-planting with green plants such as forage grass can be added appropriately. Also, on drained peatlands, ash enriched Scots pine, enabling them to grow vigorously, and improved stand nutrient status for a long period

of cultivation [46]. In poplar plantations, fertilization can significantly increase forestland organic matter, total N, and available N and, with an increase in the number of years of fertilization, there were significant differences over time [47].

Although site quality was a major factor in determining stand productivity, other silvicultural factors such as spacing (competition), edge effect, harvest treatment or vulnerability to windthrows are also critical [48,49]. Other factors that may be important include the effects of matching clones with site factors, site preparation, spacing with fertilizer application, the relationship between growth and development process of forest structure (understory diversity) as well as soil nutrient cycling on *E. urophylla × E. grandis* plantation in southern Yunnan. Thus, future research efforts can be invested in examining the effect of site and silvicultural factors on growth and appropriate reductions in planting density of high-yield groups so as to maximize productivity. In the later stages of growth, the quantitative maturity age based on the data analysis of the dominant tree of each site group will be found, and we will be able to determine the best rotation period.

## 5. Conclusions

The site types of *E. urophylla × E. grandis* plantations in southern Yunnan can be divided into 13 categories, with slope position, altitude, and soil depth as the dominant factors. High-yield *E. urophylla × E. grandis* grew best on the lower slopes, with relatively low altitude and thick soil depth composed of red soil developed by weathering of argillaceous rocks. Middle-yield *E. urophylla × E. grandis* grew relatively well on the middle and upper slopes, with slightly higher altitude and moderate soil depth, composed of red soil formed by the development of sandstone and shale and sandstone. Low-yield *E. urophylla × E. grandis* grew on the upper slopes, with higher altitude and thinner soil depth, composed of purple semi-weathered soil of purple sandy shale rocks. Total N, B, soil organic matter, Zn and Cu, and available P content had a significant effect on the growth of *E. urophylla × E. grandis* regardless of various soil types. Hence, we suggest that, apart from N, P and K, trace elements such as B, Zn and Cu should be applied to the soil with low organic matter content and pale purple soil layer. Otherwise, in reforestation after rotation cutting, planting *P. kesiya* var. *langbianensis* which is more tolerant of poor soils can be a solution.

**Supplementary Materials:** The following are available online at http://www.mdpi.com/1999-4907/11/8/871/s1, Table S1: Statistical features of dominant trees, Figure S1: Hierarchical agglomerative clustering of *E. urophylla × E. grandis* in different sites.

**Author Contributions:** H.L. wrote the manuscript and W.L. helped with the revision. H.L. and W.L. felled dominant trees and collected the soil samples. J.X. and H.L. planned and designed the research. J.X. and G.L. developed the original concept for the project, co-authored the proposal to fund the research and edited all subsequent drafts of the manuscript. All authors have read and agreed to the published version of the manuscript.

**Funding:** Project supported by the National Key Research and Development Program of China during the 13th five-year plan Period (Grant No. 2016YFD0600503).

**Acknowledgments:** The authors gratefully thank R. Pegg and J. Simpson for providing constructive advice for polishing, and at the same time thank Yunnan Yun-Jing Forestry & Pulp Mill Co., Ltd. and Weiguo State Forestry Bureau of Pu'er City, especially Ma Ning, Hu Derong, Su Guolei, Jiang Cheng, Luo Yachun, Zhang Yundong, Bai Jun and Wu Yuqiang for their painstaking assistance in standard plot survey.

**Conflicts of Interest:** Yunnan Yun-Jing Forestry & Pulp Mill Co., Ltd. is our partner in the project of "Research on the Cultivation Technology of Eucalyptus Pulpwood for High-yield Productivity". The state encourages cooperation between enterprises and research institutes, so there is no conflict of interest.

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
