# Peer review of "Site Classification of Eucalyptus urophylla × Eucalyptus grandis Plantations in China"

_forests, doi:10.3390/f11080871_

Round 1
Reviewer 1 Report
After the comments provided in my previous review authors have taken into consideration all my comments and suggestions.
However, I have minor comments:
1) The Figure 1 is not uploaded in this version. I would like to check it.
2) There some editing work to do with the tables: The Tables must to be present in only 1 page. However, mostly of them has cut (half table in one page and another in other page. Coudl you solve it to improve the editon quality of this manuscript.
Congratulations for the nice job during this review
Author Response
Dear Reviewer:
First of all, thank you for taking time out of your busy schedule to help me make valuable opinions on my thesis, which benefited me a lot. I have made corresponding changes to your suggestions. Here is the status of my changes:
1.The Figure 1 is not uploaded in this version. I would like to check it.
Yes,we done. There may be differences in the software version. If the picture still cannot be displayed, please select the picture to check the compressed picture option.
2. There some editing work to do with the tables: The Tables must to be present in only 1 page. However, mostly of them has cut (half table in one page and another in other page. Could you solve it to improve the editon quality of this manuscript.
Yes,we done.
Besides, the recent global epidemic is grim, I wish you, your family and your friends good health. Looking forward to your reply.
Kind regards,
Haifei Lu
haifeilu1122@163.com

Reviewer 2 Report
Thank you for your work.
Author Response

(The authors gave the same response as above.)

Reviewer 3 Report
Congratulations for this nice job
Author Response
Dear Reviewer:
First of all, thank you for taking time out of your busy schedule to help me make valuable opinions on my thesis, which benefited me a lot. I have made corresponding changes to your suggestions. Here is the status of my changes:
1.The Figure 1 is not uploaded in this version. I would like to check it.
Yes,we done. There may be differences in the software version. If the picture still cannot be displayed, please select the picture to check the compressed picture option.
2.There some editing work to do with the tables: The Tables must to be present in only 1 page. However, mostly of them has cut (half table in one page and another in other page. Could you solve it to improve the editon quality of this manuscript.
Yes,we done.
Besides, the recent global epidemic is grim, I wish you, your family and your friends good health. Looking forward to your reply.
Kind regards,
Haifei Lu
haifeilu1122@163.com

This manuscript is a resubmission of an earlier submission. The following is a list of the peer review reports and author responses from that submission.
Round 1
Reviewer 1 Report
- General comment :
The authors present a study about the site classification of two species of Eucaliptus in China. I appreciate the effort the author put in this research. The objective is clear and the paper structure is coherent. Therefore, the paper is very interesting and could be very useful for readers.
However, this article has some weaknesses that could be improved that I list below:
- Title is not clear
- Authors not compare Eucaliptus with other fast-growing species.
- The introduction did not present the topic context (importance of sites in the productivity), Introduction doesn’t describe the international context in other forest plantations. International context is lacking: A small paragraph to discuss how is the situation in other forest ecosystems could be great for the readers and to improve the Scopus of this paper.
-Some big mistake in the Table edition:
-Future recommendations facing the future are lacking
- In the discussion, doesn’t describes the international context. Also, in the discussion the study limitations and future research are not discussed as well as management implication.
I have also provided the following specific suggestions that would help to improve the quality of this manuscript. At this stage, I propose the authors to consider these suggestions in major revision, and request the editor not to accept the manuscript until and unless the authors make the changes. I encourage the author to incorporate these suggestions, and I want to say congratulations for this interesting study.
2. Specific comments
Tittle: - Remove the word “evaluation”. Remove “Southern Yunnan” to improve the scopus of your paper.
Abstract. Add a general background. Why this research is important? Why study the sites classification is important? Provide forest management implications.
Keywords: Keywords must to follow alphabetic order. modify it, please. Also, you could add other keywords as sustainable forest management, silviculture…
Introduction:
L39. You start talking about Eucaliptus in China. Could you provide a general background about other fast-growing species and plantations in the world. Increase the international scopus!
66-77: objective is well presented. However, you did not provided any recommendations or forest management implication in the discussion.
Methods: this section is difficult to understand because the structure.
L78: add a subsection. 2.1. study area.
Please, add a map with the study area and the study sites. Remove de decimal about temperature and precipitation, please.
L93: add a subsection. 2.2. experimental design and data compilation
L127: replace for “data analyses”
Table 1: Move it to the appendix section. This table is so big and difficult to read. It is complementary information. Check the edition: “ thickness of soil…”
Table 2. Put the units after the variables e.g. Elevation (m) and remove the unit from the body of table. Add space after “conservative slope”
Table 4. Remove Sample plot
Table 5: simplify the title of “Mean annual…” it is so long. Check edition
Table 6. I don think that R2 and F are necessary in the table. Move it to the legend.
Figure 1. It is good!However, I suggest not use abbreviations here. Replace U1 and U2 by the real name of it (index of soil…”
Figure 2: Nice
Discussion: Start the discussion with a paragraph more general. E.g. talking about the major contribution of this study. Any compartion with other species or other countries? Only China? Management implications are lacking. Add a section about it.
Table 7: replace “level” by “Level”
Author Response
Dear Reviewer:
First of all, thank you for taking time out of your busy schedule to help me make detailed reviews and valuable opinions on my thesis, which benefited me a lot. I have made corresponding changes to your suggestions. Here is the status of my changes:
I'm sorry I misspelled a word in the Abstract and misled you. Eucalyptus urophylla × Eucalyptus grandis is a hybrid of Eucalyptus urophylla as the female parent and Eucalyptus grandis as the male parent. Since 2011, Yunnan Yunjing Forestry and Pulp Mill Co., Ltd have developed the fiber raw material forest with E. urophylla × E. grandis DH32-16、DH32-22 and DH33-27 clones(E. urophylla seedlot from Alor Island, Indonesia with a selected pollen parent of E. grandis selected from a seedlot from Davies Creek near Mareeba in north Queensland, Australia)bred by the China-Australia Eucalypts Project in Dongmen Forest Farm of Guangxi.
- Specific comments
Tittle: Remove the word “evaluation”. Remove “Southern Yunnan” to improve the scopus of your paper.
Yes,I have removed the word “evaluation”and “Southern Yunnan”.
Abstract. Add a general background. Why this research is important? Why study the sites classification is important? Provide forest management implications.
Yes,I have added the Abstract in L13-19 .see revised version.
At present, the understanding of the relationship between the growth and site productivity of Eucalyptus urophylla×Eucalyptus grandis plantation in southern Yunnan is relatively shallow. In view of this, the blind expansion of cultivation areas and improper tending measures violate the principle of matching species with the site, resulting in forest degradation and low productivity. Therefore, by exploring the site types of E. urophylla×E. grandis plantation in southern Yunnan, it provides a basis for the selection of afforestation sites and scientific management in the region.
Keywords: Keywords must to follow alphabetic order. modify it, please. Also, you could add other keywords as sustainable forest management, silviculture…
Yes,I have modified the order and added other keywords “sustainable” and “silviculture”.
L39. You start talking about Eucaliptus in China. Could you provide a general background about other fast-growing species and plantations in the world. Increase the international scopus!
Yes,I have added the Introduction in L41-60. see revised version.
Forest site is the foundation of forest productivity, which is very important for forest regeneration, tree species selection, land maintenance and management. Site quality is the primary factor that determines the productivity of stand....
66-77: objective is well presented. However, you did not provided any recommendations or forest management implication in the discussion.
Yes,I have added "Implications and recommendations of forest management" to the (Discussion 4.4) in L383-412. Please see revised version.
Methods: this section is difficult to understand because the structure.
Yes,It was difficult to understand, but We revised it according to your suggestion. Please see revised version.
L78: add a subsection. 2.1. study area.
Yes,I have added a subsection. 2.1. study area in L88.
Please, add a map with the study area and the study sites. Remove de decimal about temperature and precipitation, please.
Yes, I have added a map with the study area and the study sites and remove de decimal about temperature and precipitation in L111.
L93: add a subsection. 2.2. experimental design and data compilation
Yes,I have added a subsection. 2.2. experimental design and data compilation in L113.
L127: replace for “data analyses”
Yes,I have added replaced for “data analyses” in L147.
Table 1: Move it to the appendix section. This table is so big and difficult to read. It is complementary information. Check the edition: “ thickness of soil…”
Yes,I have moved it to the appendix section and replaced "thickness of soil" with "soil thickness" .
Table 2. Put the units after the variables e.g. Elevation (m) and remove the unit from the body of table. Add space after “conservative slope”
Yes,we done.
Table 4. Remove Sample plot
Yes, we done.
Table 5: simplify the title of “Mean annual…” it is so long. Check edition
Yes, we done.
Table 6. I don think that R2 and F are necessary in the table. Move it to the legend.
Yes,we done.
Figure 1. It is good!However, I suggest not use abbreviations here. Replace U1 and U2 by the real name of it (index of soil…”
Yes,we done.
Discussion: Start the discussion with a paragraph more general. E.g. talking about the major contribution of this study. Any compartion with other species or other countries? Only China? Management implications are lacking. Add a section about it.
Yes,we have added to the Discussion in L270-280. see revised version.
Table 7: replace “level” by “Level”
Yes,we done.
Besides, the recent global epidemic is grim, I wish you, your family and your friends good health. Looking forward to your reply.
Kind regards,
Haifei Lu
Haifeilu1122@163.com
Reviewer 2 Report
Did the authors try a principal component analysis (pca)? This could be useful for combining the existing variables (independent variables) and for coming up with good correlated environmental variables.
Were the uprooted weeds left in place (as other types of forestry residues such as those coming from logging operations)?
Author Response
Dear Reviewer:
First of all, thank you for taking time out of your busy schedule to help me make valuable comments on my article, which benefited me a lot. I have made corresponding changes to your suggestions. Here is the status of my changes:
1.Did the authors try a principal component analysis (pca)? This could be useful for combining the existing variables (independent variables) and for coming up with good correlated environmental variables.
Yes, we did it used by Hierarchical Agglomerative Clustering(HAC). see revised version.
2.Were the uprooted weeds left in place (as other types of forestry residues such as those coming from logging operations)?
Not uprooted, I have replaced "uprooted" with "controlled by cutting" in L106-109.
Besides, the recent global epidemic is grim, I wish you, your family and your friends good health. Looking forward to your reply.
Kind regards,
Haifei Lu
Haifeilu1122@163.com

Reviewer 3 Report
- General comment :
The authors present a study about the site classification of two species of Eucaliptus in China. I appreciate the effort the author put in this research. The objective is clear and the paper structure is coherent. Therefore, the paper is very interesting and could be very useful for readers.
However, this article has some weaknesses that could be improved that I list below:
- Title is not clear
- Authors not compare Eucaliptus with other fast-growing species.
- The introduction did not present the topic context (importance of sites in the productivity), Introduction doesn’t describe the international context in other forest plantations. International context is lacking: A small paragraph to discuss how is the situation in other forest ecosystems could be great for the readers and to improve the Scopus of this paper.
-Some big mistake in the Table edition:
-Future recommendations facing the future are lacking
- In the discussion, doesn’t describes the international context. Also, in the discussion the study limitations and future research are not discussed as well as management implication.
I have also provided the following specific suggestions that would help to improve the quality of this manuscript. At this stage, I propose the authors to consider these suggestions in major revision, and request the editor not to accept the manuscript until and unless the authors make the changes. I encourage the author to incorporate these suggestions, and I want to say congratulations for this interesting study.
- Specific comments
Tittle: - Remove the word “evaluation”. Remove “Southern Yunnan” to improve the scopus of your paper.
Abstract. Add a general background. Why this research is important? Why study the sites classification is important? Provide forest management implications.
Keywords: Keywords must to follow alphabetic order. modify it, please. Also, you could add other keywords as sustainable forest management, silviculture…
Introduction:
L39. You start talking about Eucaliptus in China. Could you provide a general background about other fast-growing species and plantations in the world. Increase the international scopus!
66-77: objective is well presented. However, you did not provided any recommendations or forest management implication in the discussion.
Methods: this section is difficult to understand because the structure.
L78: add a subsection. 2.1. study area.
Please, add a map with the study area and the study sites. Remove de decimal about temperature and precipitation, please.
L93: add a subsection. 2.2. experimental design and data compilation
L127: replace for “data analyses”
Table 1: Move it to the appendix section. This table is so big and difficult to read. It is complementary information. Check the edition: “ thickness of soil…”
Table 2. Put the units after the variables e.g. Elevation (m) and remove the unit from the body of table. Add space after “conservative slope”
Table 4. Remove Sample plot
Table 5: simplify the title of “Mean annual…” it is so long. Check edition
Table 6. I don think that R2 and F are necessary in the table. Move it to the legend.
Figure 1. It is good!However, I suggest not use abbreviations here. Replace U1 and U2 by the real name of it (index of soil…”
Figure 2: Nice
Discussion: Start the discussion with a paragraph more general. E.g. talking about the major contribution of this study. Any compartion with other species or other countries? Only China? Management implications are lacking. Add a section about it.
Table 7: replace “level” by “Level”
Author Response
Dear Reviewer:
First of all, thank you for taking time out of your busy schedule to help me make detailed reviews and valuable opinions on my thesis, which benefited me a lot. I have made corresponding changes to your suggestions. Here is the status of my changes:
I'm sorry I misspelled a word in the Abstract and misled you. Eucalyptus urophylla × Eucalyptus grandis is a hybrid of Eucalyptus urophylla as the female parent and Eucalyptus grandis as the male parent. Since 2011, Yunnan Yunjing Forestry and Pulp Mill Co., Ltd have developed the fiber raw material forest with E. urophylla × E. grandis DH32-16、DH32-22 and DH33-27 clones(E. urophylla seedlot from Alor Island, Indonesia with a selected pollen parent of E. grandis selected from a seedlot from Davies Creek near Mareeba in north Queensland, Australia)bred by the China-Australia Eucalypts Project in Dongmen Forest Farm of Guangxi.
- Specific comments
Tittle: Remove the word “evaluation”. Remove “Southern Yunnan” to improve the scopus of your paper.
Yes,I have removed the word “evaluation”and “Southern Yunnan”.
Abstract. Add a general background. Why this research is important? Why study the sites classification is important? Provide forest management implications.
Yes,I have added the Abstract in L13-19 .see revised version.
At present, the understanding of the relationship between the growth and site productivity of Eucalyptus urophylla×Eucalyptus grandis plantation in southern Yunnan is relatively shallow. In view of this, the blind expansion of cultivation areas and improper tending measures violate the principle of matching species with the site, resulting in forest degradation and low productivity. Therefore, by exploring the site types of E. urophylla×E. grandis plantation in southern Yunnan, it provides a basis for the selection of afforestation sites and scientific management in the region.
Keywords: Keywords must to follow alphabetic order. modify it, please. Also, you could add other keywords as sustainable forest management, silviculture…
Yes,I have modified the order and added other keywords “sustainable” and “silviculture”.
L39. You start talking about Eucaliptus in China. Could you provide a general background about other fast-growing species and plantations in the world. Increase the international scopus!
Yes,I have added the Introduction in L41-60. see revised version.
Forest site is the foundation of forest productivity, which is very important for forest regeneration, tree species selection, land maintenance and management . Site quality is the primary factor that determines the productivity of stand.
66-77: objective is well presented. However, you did not provided any recommendations or forest management implication in the discussion.
Yes,I have added "Implications and recommendations of forest management" to the (Discussion 4.4) in L383-412. Please see revised version.
Methods: this section is difficult to understand because the structure.
Yes,It was difficult to understand, but We revised it according to your suggestion. Please see revised version.
L78: add a subsection. 2.1. study area.
Yes,I have added a subsection. 2.1. study area in L88.
Please, add a map with the study area and the study sites. Remove de decimal about temperature and precipitation, please.
Yes, I have added a map with the study area and the study sites and remove de decimal about temperature and precipitation in L111.
L93: add a subsection. 2.2. experimental design and data compilation
Yes,I have added a subsection. 2.2. experimental design and data compilation in L113.
L127: replace for “data analyses”
Yes,I have added replaced for “data analyses” in L147.
Table 1: Move it to the appendix section. This table is so big and difficult to read. It is complementary information. Check the edition: “ thickness of soil…”
Yes,I have moved it to the appendix section and replaced "thickness of soil" with "soil thickness" .
Table 2. Put the units after the variables e.g. Elevation (m) and remove the unit from the body of table. Add space after “conservative slope”
Yes,we done.
Table 4. Remove Sample plot
Yes, we done.
Table 5: simplify the title of “Mean annual…” it is so long. Check edition
Yes, we done.
Table 6. I don think that R2 and F are necessary in the table. Move it to the legend.
Yes,we done.
Figure 1. It is good!However, I suggest not use abbreviations here. Replace U1 and U2 by the real name of it (index of soil…”
Yes,we done.
Discussion: Start the discussion with a paragraph more general. E.g. talking about the major contribution of this study. Any compartion with other species or other countries? Only China? Management implications are lacking. Add a section about it.
Yes,we have added to the Discussion in L270-280. see revised version.
Table 7: replace “level” by “Level”
Yes,we done.
Besides, the recent global epidemic is grim, I wish you, your family and your friends good health. Looking forward to your reply.
Kind regards,
Haifei Lu
Haifeilu1122@163.com

Round 2
Reviewer 1 Report
Authors has taken into account the majority of the suggestions and comments provided.
At this stage, I propose a minor revision to consider the next two comments:
Figure 1: It is not visible! Could you add the map, please?
Discussion:
In the forest management section 4.4.:
Please talk also, other factors that could influence the yield of plantations (not only site classification). Thus, I suggest to provide some recommendation for future researches in this field. It is important to provide this reflection here: For example:
Even if site quality is a major factor in the stand productivity, other silvicultural factors as spacing (competition), edge effect, harvest treatment or vulnerability to windthrows, could influence the yield of plantations. Thus, we suggest to develop more research to study the effect of site and silvicultural factors on growth and tree-mortality to maximise the plantations productivity (Girona et al., 2016, 2019).
Montoro Girona, M., Morin, H., Lussier, J. M., & Walsh, D. (2016). Radial growth response of black spruce stands ten years after experimental shelterwoods and seed-tree cuttings in boreal forest. Forests, 7(10), 240.
Montoro Girona, M., Morin, H., Lussier, J. M., & Ruel, J. C. (2019). Post-cutting mortality following experimental silvicultural treatments in unmanaged boreal forest stands. Frontiers in Forests and Global Change, 2, 4.
Author Response
Dear Reviewer:
First of all, thank you for taking time out of your busy schedule to help me make detailed reviews and valuable opinions on my thesis, which benefited me a lot. I have made corresponding changes to your suggestions. Here is the status of my changes:
Figure 1: It is not visible! Could you add the map, please?
Yes, we done.
Discussion:
In the forest management section 4.4.:
Yes,I have added in L406-412. see revised version.
Besides, the recent global epidemic is grim, I wish you, your family and your friends good health. Looking forward to your reply.
Kind regards,
Haifei Lu
haifeilu1122@163.com

Reviewer 3 Report
Authors has taken into account the majority of the suggestions and comments provided.
At this stage, I propose a minor revision to consider the next two comments:
Figure 1: It is not visible! Could you add the map, please?
Discussion:
In the forest management section 4.4.:
Please talk also, other factors that could influence the yield of plantations (not only site classification). Thus, I suggest to provide some recommendation for future researches in this field. It is important to provide this reflection here: For example:
Even if site quality is a major factor in the stand productivity, other silvicultural factors as spacing (competition), edge effect, harvest treatment or vulnerability to windthrows, could influence the yield of plantations. Thus, we suggest to develop more research to study the effect of site and silvicultural factors on growth and tree-mortality to maximise the plantations productivity (Girona et al., 2016, 2019).
Montoro Girona, M., Morin, H., Lussier, J. M., & Walsh, D. (2016). Radial growth response of black spruce stands ten years after experimental shelterwoods and seed-tree cuttings in boreal forest. Forests, 7(10), 240.
Montoro Girona, M., Morin, H., Lussier, J. M., & Ruel, J. C. (2019). Post-cutting mortality following experimental silvicultural treatments in unmanaged boreal forest stands. Frontiers in Forests and Global Change, 2, 4.
Author Response

(The authors gave the same response as above.)
